# PLGA-Based Nanomedicine: History of Advancement and Development in Clinical Applications of Multiple Diseases

**DOI:** 10.3390/pharmaceutics14122728

**Published:** 2022-12-06

**Authors:** Hashem O. Alsaab, Fatima D. Alharbi, Alanoud S. Alhibs, Nouf B. Alanazi, Bayan Y. Alshehri, Marwa A. Saleh, Fahad S. Alshehri, Majed A. Algarni, Turki Almugaiteeb, Mohammad N. Uddin, Rami M. Alzhrani

**Affiliations:** 1Department of Pharmaceutics and Pharmaceutical Technology, Taif University, Taif 21944, Saudi Arabia; 2Department of Chemistry, College of Science, King Saud University, Riyadh 11451, Saudi Arabia; 3Department of Pharmacy, King Fahad Medical City, Riyadh 11564, Saudi Arabia; 4Pharmaceutical Organic Chemistry Department, Faculty of Pharmacy (Girls), Al-Azhar University, Nasr City, Cairo 11754, Egypt; 5Department of Pharmacology and Toxicology, College of Pharmacy, Umm Al-Qura University, Makkah 24382, Saudi Arabia; 6Department of Clinical Pharmacy, College of Pharmacy, Taif University, Taif 21944, Saudi Arabia; 7Taqnia-Research Products Development Company, Riyadh 13244, Saudi Arabia; 8College of Pharmacy, Mercer University, Atlanta, GA 31207, USA

**Keywords:** nanotechnology, nanomedicine, PLGA, polymeric nanoparticles, theranostic, drug delivery

## Abstract

Research on the use of biodegradable polymers for drug delivery has been ongoing since they were first used as bioresorbable surgical devices in the 1980s. For tissue engineering and drug delivery, biodegradable polymer poly-lactic-co-glycolic acid (PLGA) has shown enormous promise among all biomaterials. PLGA are a family of FDA-approved biodegradable polymers that are physically strong and highly biocompatible and have been extensively studied as delivery vehicles of drugs, proteins, and macromolecules such as DNA and RNA. PLGA has a wide range of erosion times and mechanical properties that can be modified. Many innovative platforms have been widely studied and created for the development of methods for the controlled delivery of PLGA. In this paper, the various manufacturing processes and characteristics that impact their breakdown and drug release are explored in depth. Besides different PLGA-based nanoparticles, preclinical and clinical applications for different diseases and the PLGA platform types and their scale-up issues will be discussed.

## 1. Introduction

Over the last decade, the number of materials used or as adjuncts in controlled drug delivery has expanded considerably [1]. Poly-lactic-co-glycolic acid (PLGA) is one of these materials. The biodegradable synthetic polymers poly-(glycolic acid) (PGA) and poly-(lactic acid) (PLA), as well as the copolymer poly-(lactic-co-glycolic acid) (PLGA), were identified as surgical sutures and monofilament in the 1960s [2]. Following the success of these polymers as surgical sutures, their usage as polymeric biomaterials has grown. Since then, the PLGA copolymer is now the most studied and widely used polymer in controlled release systems [3], and regarded as the “gold standard” of biodegradable polymers for controlled release delivery platforms [4].

Microparticles of PLGA have been widely scouted as carriers in cosmetics, food, drug delivery systems, and agriculture; they can store cargoes of both small molecules and macromolecules facilitated by hydrolysis-mediated PLGA degradation, and effectively control their release kinetics [5]. PLGA has outstanding biocompatibility and is biodegradable in the human body; also, its degradation products are CO_2_ and H_2_O, which are eliminated from the body through the Krebs cycle, and non-toxic [5]. Due to its adjustable degradation rate through variation of its monomer ratios or molecular weights, PLGA is widely used as scaffolding materials for regenerative medicine and in various biological carriers to control drug release in the field of precision therapy [5].

Moreover, PLGA has broad application prospects in gene transfer [6,7], drug delivery [8,9,10,11], tissue engineering [12,13,14], and molecular imaging [6,15,16]. For instance, biomedical applications have increased interest in polymeric nanoparticles as imaging systems [17]. One of these is PLGA nanocapsules (NCs), which turned into a substantially sensitive, MRI/photoluminescence dual-modal theranostic imaging platform for drug delivery by incorporating superparamagnetic iron oxide nanoparticles (SPIONs), as well as by combining the biocompatible and photoluminescent polyester (BPLP) into the PLGA molecular structure. The NCs adorned with SPIONs can be exploited for magnetic retention and guiding [18].

Small molecule pharmaceuticals, peptides, and proteins, such as fertility regulating hormones, anti-inflammatory medicines, growth hormones, chemotherapeutics, steroid hormones, antibiotics, cytokines, insulin, narcotic antagonists, and vaccines have been released using PLGA [19,20,21,22,23]. PLGA is very easy to manufacture into varied device morphologies such as injectable micro-/nanospheres, compared with other polymers that have been studied for controlled release [24,25].

The research and development of several cutting-edge platforms has led to the discovery of numerous strategies for the controlled distribution of PLGA. In this study, drug release and bioavailability are discussed in detail, along with the numerous production procedures and features that affect them. Preclinical and clinical applications of PLGA-based nanoparticles for various illnesses are explored, as are the many PLGA platform types and the challenge of scaling them up.

## 2. Poly(lactide-co-glycolide) (PLGA) Chemistry and History in Drug Delivery

### 2.1. History of PLGA Utilization in Drug Delivery

Biodegradable materials can be either natural or synthetic. They are metabolized in vivo, either enzymatically, non-enzymatically, or both, to yield biocompatible, toxicologically safe Byproducts removed by ordinary metabolic pathways. Polymeric synthetic microparticles have long been used to improve the bioavailability and biodistribution of both lipophilic and hydrophilic medicines [26]. As drug delivery techniques, these microparticles are becoming increasingly popular. This success can be attributed to several factors, including straightforward techniques and the potential for industrial scale-up [27]. Polymeric synthetic microparticles have many advantages as a drug delivery system, including the ability to use several administration routes and encapsulate a variety of compounds, including proteins [28]. Polymeric synthetic microparticles, in particular, can be employed for the controlled release of pharmaceuticals, with the kind of polymer and its chemical and molecular properties being adjusted [29].

The focus here is on a polyester copolymer, PLGA (poly-lactic-co-glycolic acid), which is the best-defined biomaterial due to the ability to produce controlled medication release by regulating its biodegradation. As shown in Figure 1, due to the presence of ester linkages that are degraded by hydrolysis in aqueous environments, it is controlled by polymer chemistry, such as glycoside unit content, initial molecular weight (MW) [30,31,32], stereochemistry (d and l composition) [1], or end-group functionalization [33]. PLGA’s advantages as an ideal controlled drug release agent, as shown in Figure 2, also include biocompatibility and safety (PLGA is a Food and Drug Administration (FDA)-approved substance) for human intravenous, oral, and cutaneous uses [34,35,36,37].

### 2.2. Chemistry of PLGA

Polylactic acid contains an asymmetric *α* -carbon, which is usually referred to as the D or L form in stereochemical terms but can also be referred to as the R or S form, respectively. Poly D-lactic acid (PDLA) and poly L-lactic acid (PLLA) are the enantiomeric forms of the polymer PLA. Poly D, L-lactic-co-glycolic acid (PLGA) is an acronym for poly D, L-lactic-co-glycolic acid, which has an equal ratio of D and L lactic acid types [1]. PLGA is a linear aliphatic polyester copolymer of lactic acid (CH_3_CH(OH)CO_2_H) and glycolic acid (HOCH_2_CO_2_H), which can be prepared at different ratios between its constituent monomers, lactic (LA) and glycolic acid (GA) Figure 1 [38].

## 3. PLGA-Based Nanoparticle Types and Their Scale-Up Issues

### 3.1. Types of PLGA Nanomaterials

#### 3.1.1. Polymeric Micelles

Micelles were prepared from the self-assembling amphiphilic polymers produced by the interaction between hydrophobic and hydrophilic block copolymers (e.g., PLGA and PEG di-block copolymer) [39]. Due to the biodegradability and biocompatibility of PLGA polymer, it is widely used as a hydrophobic polymer to form various kinds of block copolymers structures such as AB, BAB, or ABA in the formation of polymeric micelles [40,41]. The “core/shell” micelles developed by hydrophobic polymers define the significant properties of polymeric micelles, including drug loading or release capacity and stability. Therefore, by altering the hydrophobic block of the amphiphilic copolymer, the size of polymeric micelles can be easily regulated to exhibit a narrow size distribution, and diameters range from 10 to 100 nm [42]. The small polymeric micelle size makes it capable of selectivity targeting the tumor in cancer treatment by the enhanced permeability and retention (EPR) effect [43].

Polymeric micelles are known for their distinctive characteristics, for example, direct biological targeting, minimized toxicity, prolonged blood circulation time of drugs, stimuli-responsive properties, metabolic stability, and improved solubilization of encapsulated medicines [44]. Besides, on the in-vivo study, Yin et al. designed a PLGA micelles nanosystem with the ability to show the pH-dependent release of drugs, effectively crossing the blood-brain barrier (BBB) through micropinocytosis and lysosomal pathways [45]. In a recent study, Estupiñán et al. synthesized a novel PLGA polymeric micelles capable of encapsulating the anti-tumor antibiotic mithramycin (MTM) with high efficiency (up to 87%) and a diameter of 100–200 nm by using the emulsion/solvent evaporation method [46]. All these examples indicate the importance and applicability of utilizing PLGA for polymeric micelles as shown in Figure 2.

#### 3.1.2. Polymersomes

The liposome, which has distinctive lipid bilayers that match the cell’s plasma membrane, is an excellent and robust structure for drug administration [47]. Both hydrophilic and hydrophobic drug candidates can be incorporated into these delivery systems. As research into liposomes has progressed, several items have been put through clinical testing, and liposome-based therapies are now commonly used. Preparing liposomes with different types of lipids allows for regulation of their properties [48]. Since liposomes were discovered, their manufacture has evolved with new lipid components and processing methods. Although there are many FDA-approved liposome-based medicines on the market and more in development, clinical demands have not been satisfied [47]. Reproducibility between batches, low entrapment in some drug candidates, effective sterilizing procedures, on-shelf stability, and clinical scale-up are production obstacles. Multifunctional liposomes remain a difficulty for the industry. Hope provided by recent technological advancements supports ongoing development of liposomes and nanomedicine in general as drug delivery methods [49].

Similar to the structure of liposomes, the polymersome is a bilayer spherical nanostructure prepared from the self-assembly of amphiphilic block copolymers. However, polymersomes are more stable than liposomes due to their advantages: (i) membrane permeability and thickness that is easily adjustable based on the chain length and molecular weight of copolymers; (ii) efficiently encapsulate both hydrophobic and hydrophilic drugs; (iii) lateral diffusivity and entanglement [50,51]. Furthermore, preparation techniques of polymersomes depend on the controlled radical polymerization (CRP) methods such as atom transfer radical polymerization (ATRP), reversible addition-fragmentation chain transfer polymerization (RAFT), and ring-opening polymerization (ROP) [52]. There are two proposed mechanisms for the formation of polymersomes. In the first mechanism, the block copolymers start self-assembling into spherical micelles growing up to bilayer sheets then forming the spherical vesicles of polymersomes by reducing the edge energy. In the second mechanism, the solvent diffusion thereby reduces bending energy by raising the radius of the edges to allow the micelles to expand gradually, leading to polymersomes [53].

Stimuli-responsive polymersomes that are quite sensitive to external stimuli for smart drug release at tumor sites have been widely engineered for different therapeutic carriers such as drugs or imaging agents, nanoreactors, and other biological processes [54]. For example, a study reported applicable polymersomes composed of poly (lactide-co-glycolide-b-poly (ethylene glycol)) (PLGA-b-PEG) bilayers with grafted monoclonal antibodies (mAbs) loaded with indocyanine green (ICG). It was encapsulated at high loadings within small 77 nm polymersomes for high specificity and photoacoustic sensitivity imaging (PAI) of cancer cells. During formation of polymersomes with a water-in-oil-in-water double emulsion process, loss of ICG from the ICG aggregates was minimized via coating them with a layer of branched polyethyleneimine and via providing excess “sacrificial” ICG to adsorb at the oil−water interfaces. For 24 h in 100% fetal bovine serum, the encapsulated aggregates were protected against dissociation via the polymersome shell after which the polymersomes biodegraded and the aggregates dissociated to ICG monomers [55]. In recent studies, well-designed drug delivery systems were constructed via an adapted double emulsification method for preparing the DOX-loaded PLGA nanoparticles (DOX@PLGA), followed via the modification of different shells using the amino-terminated polymers. They bound with negative charged carboxyl of PLGA nanoparticles by electrostatic interaction. For folic acid receptors overexpressed on cell membrane of tumors, folate grafted amine poly (ethylene glycol) (NH2-PEG-FA) was utilized to target tumors and quaternary chitosan (QCS) was introduced for prolonging circulation. To shield the negative charge of nanoparticles and enhance the interaction of nanoparticles with the cell membranes, the Poly (allylamine hydrochloride) (PAH) was introduced. Also, the developed degradable core–shell polymersomes were able to release DOX in a controlled and pH-dependent manner, in which significantly facilitated drug release was observed at a mildly acidic pH of 5.0 compared with physiological pH (pH 7.4) [56].

#### 3.1.3. Lipid Nanoparticles

The lipid-polymer nanosystem consists of a lipid shell surrounding the polymeric core, representing a favorable characteristic for drug delivery application due to its high stability and encapsulation efficiency [57]. PLGA is the most commonly used polymer as the inner polymeric core in preparing lipid-polymer hybrid nanoparticles [58]. The designing of lipid-PLGA nanoparticles depends on the optimal therapeutic effect. Additionally, there are several types of lipid-PLGA nanoparticles, such as PLGA core-lipid and lipid core-PLGA. On the first one, the lipid shell embeds the therapeutic agents surrounding the PLGA core. Otherwise, in the lipid core-PLGA, the PLGA polymer encapsulates the lipids core [8]. Different engineering methods as shown in Table 1 describe the fabrication of lipid-PLGA nanoparticles, such as the single-step method and two-step method. The two-step method is a conventional process where the performed lipid liposome is absorbed on the surface of the performed PLGA nanoparticles by electrostatic interactions. In the single-step method, the PLGA polymer and lipids are dissolved in a suitable solvent under the same conditions and tend to self-assemble the lipid around the PLGA nanoparticles.

Several techniques for each process depend on various factors as shown in Figure 2 such as size, shape, and nature of drug incorporation with the engineered nanoparticles [8,61]. For instance, García-García et al. used single-step nanoprecipitation techniques to synthesize lipid-PLGA nanoparticles with variable surface charges, incorporating GapmeRs single-strand antisense oligonucleotides for osteoporosis therapy by adjusting the lipid composition [62]. Lipid-PLGA nanoparticles loaded with the Paclitaxel drug were also successfully synthesized via a single-step nanoprecipitation technique in a 150–400 nm size range [63]. Moreover, in another study, Maghrebi et al. engineered PLGA-lipids as a drug carrier system loaded with antibiotics to treat intracellular pathogens by using a two-step method through the process of spray drying [64]. Currently, there are many ongoing preclinical studies utilizing these platforms for multiple diseases.

### 3.2. PLGA Systems Scale-Up Productions: Challenges and Efforts

Scaling up production without affecting formulation requirements obtained at the lab scale is one of the major challenges in the clinical and commercial creation of sub-micron polymeric particle formulations. As shown in Table 1, several methods for producing sub-micron PLGA particles on a lab-scale have been developed, and they all use emulsion-based batch techniques. Because of the low cost of equipment and ease of use, emulsification through direct sonication with a transducer probe is one of the most common methods for forming PLGA particles. However, when this process is scaled up to industrial batch sizes, it can induce changes in particle properties, including drug release profiles. Continuous processes have the advantage of allowing output to be stopped at the desired scale without adjusting the process parameters. The high-shear mixing method uses extreme shear forces to minimize mixing times in processes involving immiscible fluids formed into emulsions [65]. Since PLGA tends to accumulate in the hydrophobic polydimethylsiloxane (PDMS) channels and eventually clogs them, large-scale development of PLGA-based NPs for clinical studies is limited. However, since PDMS can only withstand a small amount of high pressure, further elevation of the flow rate is limited and thus cannot meet the requirements of a large-scale clinical trial. However, some materials, like glass capillaries and polytetrafluoroethylene, can be used to make microfluidic chips instead of PDMS. They can withstand intense pressure, allowing for high throughput of PLGA NPs [66].

A new method for preparing PLGA microspheres on a pilot scale has recently been developed as shown in Figure 3. The technique has many benefits, including a high yield, minimal post-process handling, and a fast operation time [67]. Emulsification in a packed-bed column is the mechanism. A continuous phase runs through gaps between beads (50–1000 m) filled inside the column, forming emulsions. The emulsions repeatedly travel through the openings, narrowing the size distribution. The obtained microspheres are in the 10–100 m size range, and the span value (a size distribution index) estimated from the submitted data is around 0.6 (indicating narrower size distribution). The equipment’s flow rate is stated to range from 0.0001 to 100 L/min. Furthermore, each batch’s production scale might range from dozens to hundreds of kilograms.

Edge Therapeutics^®^, for example, has produced PLGA microspheres encapsulating nimodipine for the treatment of aneurysmal subarachnoid hemorrhage using this approach [68]. For a certain therapeutic release profile, the manufacturer also developed a program called “Precisa” that uses a specific combination of polymers. This is done by entrapping the treatment in a biodegradable and biocompatible matrix of clinically validated polymers. PLGA, a polymer utilized in dissolvable sutures since the 1970s, is the underlying material of “Precisa”. Even when employed intracranially, the biodegradability of PLGA and its low toxicity to humans make it an ideal matrix for long-term drug administration. When the polymer is injected into a patient, the therapeutic agent on its surface is rapidly released, resulting in high initial concentrations of the drug. Polymer-based microparticles are then dissolved in lactic acid, a naturally occurring chemical, to deliver therapeutics in accordance with the intended release profile of the microparticles’ therapeutic agents.

### 3.3. Efficacy and Safety Assessment: Ways of Clinical Translations

Pilot studies in large animal models, where protection and effectiveness can be investigated over long periods, are needed to translate initial successes in small laboratory animals into clinics [69]. Aside from optimizing formulation and process factors, the safety and efficacy by design method should incorporate the biological response (nano-bio interactions) to tailor physicochemical characteristics for effective tumor targeting [70]. For example, anticancer combination therapy has been shown to be a more successful treatment technique than a single drug delivery method. Thus, recent innovative therapy has the ability to minimize side effects, improve medication effectiveness, and resolve multidrug resistance, which may be a significant barrier to anticancer chemotherapy success. To create lipid-coated nanoparticles, mTHPC liposomes were coated onto the chosen THP nanoparticles based on their physicochemical profiles (LCNPs), as presented in Figure 4. The histopathological investigation of the vital organs showed no apparent signs of toxicity, implying that the PLGA lipid-polymer hybrid system is healthy and efficient. It entails determining the hemolytic ability of drug formulations in the presence of blood components, which will decide their therapeutic effectiveness and in vivo fate [71]. Anti-tumor drugs encapsulated in PLGA NPs can thus not only increase anti-tumor effectiveness but also greatly minimize side effects [72]. There has been a huge effort in investigating different PLGA platforms and evaluating their safety and efficacy [8,73,74,75,76].

### 3.4. Biodistribution Studies of PLGA Nanomedicine Formulations

Systemic administration of nanoformulation is widely used to study the biodistribution, disease targeting, and therapeutic effectiveness of NPs-based drug delivery systems as shown in Table 2. Additionally, near-infrared imaging allows for time-resolved biodistribution studies. Fluorescent dyes emit near-infrared light, enabling spatiotemporal analysis in conjunction with optical tomography [77]. Thus, PLGA-NPs could be used to shield the medication from gastric degradation while also reducing its absorption by mucosae villi or Peyer’s patches and subsequent release into the bloodstream [78]. PLGA is characterized by its in vivo biocompatibility, biosafety, and biodegradability, which attract researchers to use it as a carrier in many diseases such as cancer [79]. PLGA nanoparticles play a role in enhancing the bioavailability of the encapsulated payload and minimizing the premature degradation in the biological systems [8]. Despite the advantages of PLGA as a carrier, lack of specificity in cells and protein binding is the main PLGA drawback that minimizes its accumulation in the target tissue [80,81]. Thus, modifying PLGA nanoparticles, as shown in Figure 5, is essential to improve the drug delivery properties [79]. Maksimenko et al. coated PLGA nanoparticles with poloxamer 188 to improve the brain delivery of doxorubicin. The study showed that the efficiency of the coated PLGA nanoparticles and accumulated drug delivery showed higher efficacy in the target site compared with free doxorubicin [82]. Moreover, Partikel et al. showed that pegylation of PLGA nanoparticles reduced the protein bounding and immune systems recognition, which enhanced the biodistribution of the nanoparticles within the biological systems [83].

## 4. PLGA as a Platform for Drug Delivery Systems

The US Food and Drug Administration (FDA) approved using the PLGA for various pharmaceutical applications. In 1989, the first FDA-approved drug delivery system, Lupron^®^ Depot, was released based on a biodegradable polymer. Lupron^®^, Leuprolide is contained within PLGA microspheres and used as a depot for the treatment of prostate cancer [90]. Medication release from this biodegradable formulation may be modulated by adjusting the biodegradation of PLGA, leading to a prolonged drug release profile that decreases harmful side effects and improves patient compliance [91]. There are many other US FDA-approved PLGA-based marketed products such as Zoladex Depot ^®^(AstraZeneca UK Limited), Sandostatin^®^ LAR, Suprecur^®^ MP, and many other more as listed in Table 3.

Additionally, to enhance the administration of hydrophobic medicines, PLGA, a biocompatible and biodegradable polymer, is frequently utilized in clinical and preclinical settings for nanoparticle production (as shown in Figure 6 and Table 2) [92]. Zhang et al. developed curcumin (CUR)-encapsulated chitosan-coated poly (lactic-co-glycolic acid) nanoparticles (CUR-CS-PLGA-NPs) and hydroxypropyl-β-cyclodextrin-encapsulated CUR complexes (CUR/HP-β-CD inclusion complexes) for Alzheimer’s disease. In vitro studies indicated that both CUR-CS-PLGA-NPs and CUR/HP-β-CD inclusion complexes were very stable over two months of storage. Moreover, the study revealed that CUR-CS-PLGA-NPs and CUR/HP-β-CD inclusion complexes could minimize the toxicity of CUR and show excellent antioxidant and anti-inflammatory activities [93].

Furthermore, Upadhyay et al. studied the role of adding targeting ligand to nanoparticles. They found that PLGA encapsulated silymarin that functionalized with lactobionic acid (LA) (LA-PLGA-Sil) exhibited better targetability, cellular distribution, and toxicity against the liver cancer HepG2 cell line compared with non-targeted formulation PLGA encapsulated silymarin (PLGA-Sil) [95].

In another study, Hu et al. has successfully fabricated a Fe^III^-TA complex-modified PLGA nanoparticle platform for the tumor-targeted delivery of Doxorubicin (DOX) to treat breast cancer. The nanoformulation successfully decreased premature release of chemotherapy drugs during systemic administration and enhanced pH-responsive release in the tumor microenvironment. At pH 5.0, the cumulative drug release rate was 40% greater than at pH 7.4 [96]. Additionally, nanoparticles made from PLGA 503H and the lower molecular weight PLGA 2300 loaded with trimethoprim (TMP) for urinary tract infections (UTI) by Brauner et al. were tested for their ability to adhere to cells, and the results were significant. The nanoformulation surface was modified by wheat germ agglutinin (WGA) to circumvent the short dwelling time in the bladder. In this study, both PLGA types showed almost 70% adhesion capability to the cell surface of SV-HUC monolayers within 30 min contact time [97].

Zhang et al. developed a PLGA nanoparticle, which encapsulated the antigenic peptide HPV16 E744–62 to treat HPV-associated tumors [73]. This system provides some distinct advantages, such as minimizing peptides’ degradation, enhancing the peptides’ residence time, promoting the uptake efficiency by an antigen-presenting cell (APCs), and enhancing peptide migration/accumulation into lymph nodes leading to the presence of more APCs. The results showed that adenosine triphosphate (ATP) is a new and potent vaccine adjuvant; thus, encapsulating ATP within PLGA nanoparticles elicits robust anti-tumor cellular immunity [73,97]. Far et al. designed PLGA NPs encapsulated mometasone furoate (MF) for sustained drug release using the nanoprecipitation method. This delivery system exhibited adequate physicochemical properties, high drug encapsulation efficiency, and loading, which makes it suitable for treating chronic rhinosinusitis [98].

## 5. PLGA Nanomedicine Formulations as a Platform for Theranostic: Imaging and Biodistribution Studies

Integration of diagnostic and therapeutic chemicals into a flexible nanocarrier is seen as a potential cancer therapy strategy due to the expectation that it would increase the anti-tumor activity and decrease the adverse effects of traditional chemotherapy. Using the double emulsion solvent evaporation approach (W/O/W), which is linked to changes in bovine serum albumin (BSA), Shen et al. constructed a PLGA-based theranostic nanoplatform as shown in Figure 7. Co-delivery of the near-infrared (NIR) dye indocyanine green (ICG) and the chemotherapeutic medication doxorubicin (Dox) (denoted as IDPNs) for dual-modality imaging-guided chemo-photothermal combination cancer treatment was accomplished with this delivery method. A minimal quantity of tumor accumulation was attained by the free ICG. Additionally, the IDPNs accumulated mostly in the tumor, kidneys, and liver 24 h post-injection, producing a robust fluorescence signal in the removed tumors [99].

Furthermore, Zhang et al. prepared PLGA nanocapsules (NCs) as a multimodal theranostic delivery system for in vivo/animal drug delivery. They outlined the chemical synthesis techniques for covalently labeling the PLGA with bio-compatible small molecule fluorophores or radioligands. Bovine serum albumin (BSA) was used as a model protein, while superparamagnetic iron oxide nanoparticles (SPIONs) were used as an MRI contrast agent. ^89^Zr-labeling was used as a radio imaging probe, and fluram and cyanine 7.5 (Cy7.5) were used as fluorescent probes in the blue and near-infrared (NIR) wavelengths, respectively. Quantitative region of interest (ROI) analysis reveals a reduction of 55% in signal intensity (SI) on T2W images and 26% in T2 relaxation time in the liver. In comparison, the kidney showed only a slight decrease in signal strength. After in vivo MRI, ferrous iron was detected in liver slices stained with Prussian blue, confirming the existence of NCs. Cell viability was not affected by exposure to NCs ranging in concentration from 25 to 100 g mL^−1^ for 48 h following overnight incubation at room temperature at a concentration of mg mL^−1^ [16].

Additionally, nanocapsules composed of a PLGA-polymer matrix coated with Fe/FeO core-shell nanocrystals and co-loaded with a chemotherapy drug and photothermal agent were used in a study by Wang et al., which also made use of near-infrared light and the tumor microenvironment (TME), dual responsiveness, and size-switchability. ICG@Fe/FeO-PPP-FA nanocapsules were found to have low cytotoxicity, as 95% cell viability was observed after being incubated. It was found that ICG@Fe/FeO-PPP-FA nanocapsules accumulated much more than ICG@Fe/FeO-PPP nanocapsules at the tumor site, as the fluorescence signals were observed and strong in the tumor and liver but not in the main organs (spleen, lung, heart, and kidneys) after 48 h of intravenous injection (20 mg/kg, 200 mL). The tumor site MRI signal intensity of DOX-ICG@Fe/FeO-PPP-FA nanocapsules was 85 a.u. with and 65 a.u. without laser irradiation (808 nm, 0.3 W cm^−2^, 5 min) after 24 h. Therefore, laser-triggered shrinking of DOX-ICG@Fe/FeO-PPP nanocapsules proved useful for the deep tumor tissue penetration of nanocapsules [100].

In another study, the PLGA-NPs were loaded with a recombinant human VEGF-A165 (the vascular endothelial growth factor, rhVEGF) analog via the 1-Ethyl-3-[3-dimethylaminopropyl] carbodiimide hydrochloride/N-hydroxysuccinimide (EDC/NHS) covalent coupling method as a new theranostic technology for tumor diagnosis and therapy by Varani et al. as shown in Figure 8. Mice that were injected with VEGF-PLGA-NPs represented elevated tumor uptake and higher target-to- muscle (T/M) ratio in comparison with PLGA-NPs, where the tumor uptake and T/M ratio of VEGF-PLGA-NPs were 39.83 ± 7.17 and 7.90 ± 1.61, while for PLGA-NPs they were 29.95 ± 1.92 and 4.49 ± 0.54, respectively [101].

Chen et al. created a new form of multiporous lipid/PLGA hybrid microbubbles (lipid/PLGA MBs) that resolved the challenge of microbubbles as imaging agents and drug carriers. The study utilized doxorubicin-loaded nanoparticles (Dox-lipid/PLGA MBs) as a model drug delivery system. The liver ultrasonography contrast signal was low before bubble injection but improved rapidly after lipid/PLGA MBs were injected. The ultrasonic assessment performed in vivo by these MBs is highly efficient. This technique offers novel understanding of the biological action of medications used to treat tumors and other disorders by regulated drug release, as well as tracking their locations within the body [102].

## 6. PLGA Nanomedicine Platforms for Different Diseases

Due to low toxicity, biocompatibility, controlled and sustained-release properties with tissue and cells, PLGA has been utilized in clinical drug delivery systems as one of the most effective biodegradable polymeric nanoparticles (NPs) as shown in Table 4 [11,103,104,105]. In the next subsections, we highlight some of the most important usages of PLGA for different diseases and purposes.

### 6.1. Cancers

Nanoparticles can target cancer cells through passive or active targets. Drug resistance, low intra-tumoral accumulation, and non-specific cytotoxicity are the problems most associated with chemotherapeutic agents; thus, using PLGA as a delivery system has gained significant attention due to its outstanding properties [106]—for example, the blood-brain barrier (BBB), which is one of the major limitations facing chemotherapeutic drugs delivery to the brain in the case of malignant gliomas. The anticancer agent paclitaxel has limited BBB permeability, but when combined with a nonsteroidal anti-inflammatory drug that has anticancer activity such as R-Flurbiprofen and carried by PLGA NPs, the use of anti-inflammatory and anticancer drugs may provide additional anti-tumor activity; this conjunction has also decreased inflammation in the peri-tumoral area [107]. Moreover, 5-Fluorouracil (5-FU) is extensively used as first-line chemotherapy for colon cancer. Yet, systemic toxic effects and low drug uptake limit its use. Several novel drug delivery systems as dendrimers, liposomes, and polymeric nanoparticles (NPs) have been reported to overcome these limitations. PLGA is widely used as a delivery system for multiple drugs, including 5-FU, due to biodegradability and biocompatibility. Therefore, another technique used to prepare NPs is the double emulsion method; this has been used to develop PHBV/PLGA NPs as a novel combination drug delivery system. 5-FU loaded PHBV/PLGA NPs induced considerably higher cell death than free 5-FU in colon cancer [108].

However, the effectiveness of chemotherapy in the case of glioblastoma multiforme (GBM) is primarily limited due to scant brain delivery of most therapeutics across the blood-brain barrier (BBB) [109,110]. A promising strategy that may offer a solution for this problem is drug delivery by nanoparticles. However, one of the most effective anti-tumor drugs that could be used in combination or alone is doxorubicin; it has been used as the first-line therapy in many cancers. On the other hand, it has been used as a drug of choice in drug delivery-related studies aimed to overcome tumor resistance. PLGA nanoparticles loaded-doxorubicin increased survival time, and has a significant inhibition on tumor growth, and a very considerable anti-tumor effect against the intracranial glioblastoma in the rats animal model [82].

Another example is epidermal growth factor receptor (EGFR), which is highly expressed in pancreatic cancer (PC). EGFR inhibitor’s use alone was proven ineffective in clinical trials due to a cellular feedback mechanism that takes over therapeutic resistance to single targeting of EGFR [111]. Specifically, the signal transducer and activator of transcription 3 (STAT3) when receiving an EGFR is overactivated and considered to be highly involved in the resistance and failure of EGFR inhibitor treatment. For that, PLGA NPs were co-loaded with Erlotinib (ERL), an EGFR inhibitor and one of the first-generation approved for lung and pancreatic cancer treatment [112]. Furthermore, Inula helenium Alantolactone (ALA) was confirmed to possess the STAT3 inhibition property. Firstly, for the best therapeutic outcome, the ERL and ALA ratio was screened. Then, PLGA NPs co-loading ALA and ERL were characterized and optimized. A nanoplatform to co-deliver ALA and ER showed anti-migration and antiproliferation effects and an enhancement in cell-killing resulting from increased cellular uptake in PC cells. However, this co-delivery markedly inhibits both STAT3 and EGFR signaling pathways [113]. There are many examples [114,115,116] that can be highlighted for the utilization of PLGA in cancer research as shown previously in Table 2.

### 6.2. Neurological Diseases

Multiple clinical conditions are often associated with neuropathic pain, including patients undergoing chemotherapy courses and diabetic neuropathy. Therefore, antiepileptic drugs (AED) help detract neuronal excitability as they have common pathophysiology for epilepsy and neuropathy. Lamotrigine (LTG), approved AED, is widely used as a first-line treatment for neuropathic pain [117]. With this AED, a modified nanoprecipitation method was used to prepare LTG-PLGA-NPs. Studies mentioned PLGA as an appropriate carrier system for lamotrigine for neuropathic pain using the intra-nasal route [118]. Moreover, the modified nanoprecipitation method was used to prepare nanoparticles of baclofen (Bcf-PLGA-NPs); the allocation of Bcf intranasally provides direct transmucosal absorption to the brain passing blood-brain barrier (BBB), giving an appropriate administration option, and rapid and early onset of action [119].

On the other hand, the neurodegenerative disorder mainly recognized by b-amyloid deposit known as Alzheimer’s disease (AD) has no curative treatments. Yet, curcumin (Cur), with its anti-inflammatory, antioxidant, and anti-amyloid properties has been proven to have future use in Alzheimer’s disease. B6 peptide enhances the BBB permeability conjugated with a novel brain-target nanoparticle poly(lactide-co-glycolide)-block-poly(ethylene glycol) (PLGA-PEG) to increase the bioavailability. It is loaded with Cur (PLGA-PEG-B6/Cur), which has a promising property in relieving tauopathy and beta-amyloidosis. These results indicate that PLGAPEG-B6 can enhance the delivery of nanoparticles to the brain, and PLGA-PEG-B6/Cur nanoparticles may be used as a promising therapeutic approach for treating AD in the future [120]. The construction of a rational medication therapy for AD has had limited success despite extensive research into nano-based drug delivery [121].

### 6.3. Cardiovascular Diseases

Numerous patients suffer from cardiovascular diseases; a recognized significant cause of death worldwide [122,123]. Further, atherosclerotic disease of the carotid veins occurs in the aging population. Thus, stabilin-2 peptide (S2P) was highly expressed on smooth muscle, endothelial cells, and atherosclerotic plaques. For this reason, targeting agents for atherosclerosis with localization of nanoparticles containing S2P peptide has been brought up. Therefore, the focus on inhibition of platelet adhesion, aggregation, and activation is necessary as one of the efficient therapeutic approaches [124,125]. Imatinib as a platelet-derived growth factor receptor (PDGFR) inhibitor has very low water solubility. This low water solubility was overcome by using peptide conjugated nanoparticles as a drug delivery system for atherosclerotic disease. Thus, Imatinib was encapsulated in PLGA nanoparticles which are conjugated to maleimide PEG [126].

Furthermore, using PLGA as a drug carrier for serum lipid-lowering drugs such as statins is a valid way to enhance statins’ efficiency [127,128]. Hydrophobic drugs with bioabsorbable PLGA polymer were taken up by various cells such as vascular smooth muscle cells, monocytes, and endothelial cells. In addition to their serum lipid-lowering ability, statins have pleiotropic effects and improve collateral circulation in the ischemic heart [129]. Therefore, using a bioabsorbable polymer such as (PLGA)-nanoparticles loaded with simvastatin (SimNPs) reduced the infarct site area and improved cardiac function in the treated group with (SiMNPs) [130].

### 6.4. Infectious Diseases

Sepsis occurs when an invasion of pathogens in the host is triggered by an immune disorder; it is complex with very high mortality and morbidity. It is challenging to simultaneously have an effective delivery of immune-modulating and anti-infection drugs. Therefore, prepared PLGA nanoparticles with good degradability and biocompatibility were co-loaded with anti-inflammatory immunosuppressant Tacrolimus (TAC) and broad-spectrum antibiotic Sparfloxacin (SFX). The NPs have outstanding antibacterial activity on both Gram-negative and Gram-positive bacteria and can effectively reduce the immune response and inflammation in mice with acute lung infection [131].

A fatal tropical disease known as Visceral leishmaniasis (VL) is caused by the parasite Leishmania donovani, transmitted to humans by the bite of an infected sandfly. The aminoglycoside antibiotic paromomycin (PM) has significant antileishmanial activity. However, due to the lack of oral bioavailability and decreased permeability across the macrophage’s cell membrane, the full therapeutic potential of PM is endangered by its decreased accumulation inside the macrophages. For this reason, PLGA nanoparticles were prepared to encapsulate the drug PM and coated with Mannosylated thiolated chitosan (MTC). L. donovani amastigotes were effectively inhibited by the mannosylated PLGA nanoparticles in affected macrophages compared with the pure drug. Therefore, PM-loaded MTC-PLGA nanoparticles successfully promoted anti-leishmanial activity [132].

In another example, in a pathological condition observed after surgery associated with microbial growth and biofilm formation known as surgical site infection (SSI), Oral consumption of Vancomycin, and aminoglycosides such as gentamycin sulphate (GS) is the preferred alternative for preventing post-operative SSI. Thus, PLGA was chosen as a polymer for the preparation of PLGA polymeric nanoparticles (PNPs). Moreover, Gum Kondagogu (GKK) is an exudate from the rind of Cochlospermum Gossypium, and it is composed of galactose, glucose, arabinose, and non-toxic polysaccharide. Therefore, GS loaded PLGA-GKK encapsulated PNPs were elicited for the treatment of SSI at the localized site [133]. Additionally, synthesis of the PLGA/Ag2O NPs composite was performed by a low-temperature technique established by Smirnova et al. [134]. The nanocomposite produced encouraging findings, suggesting it may be used to create materials with both potent antibacterial activity and great biocompatibility with human cells. Such materials have potential uses in surgical procedures, especially in prosthetics [134].

### 6.5. Other Diseases

PLGA has been utilized widely in improving the efficacy of many drugs used in treating or managing multiple diseases such as ophthalmic delivery systems, periodontal regeneration, and chronic obstructive pulmonary diseases delivery systems [135,136].

#### 6.5.1. Ophthalmic Delivery Systems

The anatomy of the eye plays a major role in restricting the amount of drugs that reach the site of the affected tissue [136]. Different barriers in the anterior portion of the eye limit the penetration of the drugs, which results in poor drug bioavailability [137]. The tear turnover is one of the eye mechanisms in which topical drugs are diluted. Another barrier is the cornea, which is composed of several layers such as the corneal epithelium, Bowman’s layer, corneal stroma, Descemet’s membrane, and corneal endothelium [138]. The corneal epithelium, which is lipophilic in nature, limits the penetration of hydrophilic drugs. Corneal stroma, which is composed of collagen fiber, is another barrier that limits not only foreign bodies, but drug absorption. Furthermore, corneal endothelium is another barrier that works as a barrier between the cornea and aqueous humor [138,139]. Besides the anterior eye segment barriers, the posterior eye segment is another barrier that limits drug absorption. The vitreous body, which is one component of the posterior segment, is a gel-like structure that functions as a barrier that limits the movement of drugs from the vitreous humor to the retina [136,137]. Thus, it is important to develop several strategies to enhance the efficiency and bioavailability of current eye drugs by implementing drug delivery systems technology. Using PLGA as a nanocarrier is one of the ways it has been utilized to improve the overall drug efficacy and bioavailability [136].

Drug therapy for the back of the eye is often administered intravitreally [140]. However, prominent downsides of this treatment include the drug’s short residence duration and high clearance, irritation caused by numerous injections, and the potential of vision impairment. In light of these constraints, researchers have been working to develop intraocular drug delivery methods that can address at least some of the issues just discussed. In this way, PLGA and PEG monomer units form a class of amphiphilic water-soluble polymers known as PLGA-PEG-PLGA triblock copolymers [141]. Micelle-like structures, with a high hydrophobic core (PLGA) and an encircling corona-like structure built of PEG tails, can be formed from PLGA-PEG-PLGA copolymers in aqueous solution [142]. Though the details are modifiable, the micelles are typically well-separated and distributed at room temperature, giving the solution a sol state. However, the micelles grow in size and begin to aggregate at higher temperatures, transitioning into the gel state and thusly producing a hydrogel that responds to temperature. It has been hypothesized that polymer precipitation and PEG chain dehydration contribute to micelle disintegration at elevated temperatures [143]. Many other studies suggested that PLGA-PEG copolymers can be used as a delivery platform to lengthen the amount of time that active compounds spend in the eye’s back chamber, enabling for the development of treatments that don’t require as frequent administration [75,144,145,146].

#### 6.5.2. Periodontal Regeneration

Biocompatible and biodegradable PLGA polymer has been studied for periodontal regeneration [147]. A fibronectin-functionalized electrospun PLGA scaffold [148] was created to increase periodontal ligament cell adhesion. In vitro experiments showed better cell adherence to fibronectin-functionalized scaffolds than unfunctionalized ones. Moreover, fibronectin permitted a more homogenous cell adhesion over the whole scaffold surface, demonstrating a unique extracellular matrix deposition and acquired cell migratory capacity. Fibronectin-functionalized PLGA fibers provide a superior scaffold for periodontal regeneration than non-functionalized ones [148]. PLGA hydrogels for periodontitis were also studied. An injectable scaffold consisting of PLGA and hydroxypropyl methylcellulose (HPMC) loaded with chlorhexidine was developed [149]. Its syringeability, textural profile, and swelling/shrinking characteristics were compared with two gels on the market: Parocline^®^ (minocycline gel) and Chlo-site^®^ (chlorhexidine gel). The researched new hydrogel had better syringeability than Parocline^®^ and Chlo-site^®^, although its textural characteristics were midway between the two gels on the market. These studies showed PLGA/hydroxypropyl methylcellulose hydrogels had better physicochemical qualities for periodontal use. HPMC improved gel property, giving more robust and adequate physical support for in vivo periodontal application [149]. Additionally, the multilayered PLGA-calcium phosphate scaffold was presented by Reis et al. [150]. This scaffold was strong and flexible, adjusting to its new environment with ease, and its connected macroporosity allowed for blood reabsorption upon implantation. This study set out to create a bilayered construct for periodontal regeneration in vivo. The 30-, 60-, 90-day, and 12-month durability of this scaffold was evaluated in canine subjects. Histological examination of the treated animals showed cementum, alveolar bone, and periodontal ligament regeneration. In comparison with absorbable membrane, the rigidity of this construct decreased collapse, promoting regeneration of wounded periodontal tissue [150].

#### 6.5.3. Chronic Obstructive Pulmonary Diseases (COPD)

Pulmonary emphysema, the form of COPD that causes chronic breathing difficulties, is a major cause of global mortality [151]. An overabundance of free radicals and chronic inflammation are responsible for chronic obstructive pulmonary disease (COPD). Substances with anti-inflammatory, antioxidant, and corticosteroid properties are toxic, need high doses, and have serious side effects [152]. Due to their adaptability to specific microenvironments in diseased tissues and low toxicity, nanomaterial-conjugated medications show promise in treating a wide range of respiratory illnesses. The pharmacological effects of small RNA molecules and drug conjugates designed for the treatment of chronic respiratory illnesses can be enhanced by loading them onto PLGA NP [153]. Therapeutic applications for PLGA NPs in the treatment of a variety of respiratory disorders are promising. In terms of developing PLGA miRNAs for clinical uses in COPD, insufficient advancement has been made. The fast breakdown of miRNA after being administered to humans may be the underlying cause [154]. Polymer’s immunogenicity is a further explanation. Loss of effectiveness of polymer-coated medicinal compounds has been linked to the existence of antipolymer antibodies [155]. Improved targeting efficiency and decreased off-target effects can only be achieved by adjusting the PLGA NP encapsulation of miRNAs [156].

Numerous pharmacologically active small compounds have been identified for the treatment of COPD; however, their limited permeability across the mucus lining prevents their widespread clinical application [157,158]. To counteract this, the anti-inflammatory medication ibuprofen was conjugated with the PLGA-PEG NP to specifically target the neutrophil-mediated inflammatory response in COPD [159]. Ibuprofen’s anti-inflammatory properties and its ability to reduce lung harm caused by lipopolysaccharide (LPS) and cigarette smoke (CS) show promise for its use in clinical settings. Additionally, the generation of reactive oxygen species (ROS) and interleukin-8 (IL-8) in COPD is suppressed when the antioxidant 1,3-di [5-(N-methylene-pyridinium-4-yl)-10,15,20-triphenylporphynato manganese]-benzene tetrachloride (MnPD) is coupled with PLA NP [160].

When it comes to treating respiratory conditions, nanomedicines conjugated with PLGA are effective, safe, and convenient [161]. Despite this, relatively little research has been reported in preclinical settings on the use of PLGA-conjugated medication to fight COPD. The effective treatment of COPD necessitates the rapid development of innovative medications, such as enhanced PLGA-drug conjugates, that provide more efficacy with reduced toxicity. More study is needed to develop a formulation for drug-loaded PLGA NP and track the rate at which the medication is released in the respiratory system. The development of PLGA-drug conjugates for the treatment of COPD will benefit from this.

## 7. Conclusions and Future Directions

Since their discovery, PLGA-lipid hybrid nanoparticles have garnered considerable interest as promising multipurpose carriers for a wide variety of therapeutics, leading to a wide variety of successes in the design of new drug delivery systems and making a particularly noticeable impact in the field of cancer therapy. When compared to other drug delivery formulations, PLGA-lipid hybrids have shown that they are better in terms of physicochemical characteristics, shape, and biological activity. While these results are impressive, much more work has to be done to get a thorough comprehension of certain technological, biological, and industrial elements that may lead to novel options for the creation of safe and powerful carriers with medical applications. There are two main obstacles to overcome when designing a hybrid formulation: first, a thorough understanding of the interactions of the hybrid nanocarrier with cells is required to predict the potential toxicity issues and ensure its safe clinical applications; and second, a comprehensive consideration of essential characteristics of each constituent that forms the hybrid system is required for successful design.

Some PLGA-lipid hybrid platforms are only manufactured in small quantities because of the intricacy of the engineering technique, making scale-up problematic. Cost-effectiveness and repeatability favor single-step procedures over two-step ones. Single-step methods are not always possible for PLGA-lipid hybrids, necessitating separate optimization. Due to the complexity of biological environments, trustworthy and highly reproducible hybrid carriers are needed for clinical studies. Increasing the complexity of hybrid formulations may lead to an increase in expenses, complicating subsequent experiments. Despite advances in building increasingly complicated hybrid carriers, multicomponent architectures can hinder translation into a pharmaceutical and clinical program.

As a result, the use and investigation of clinical PLGA nanocarriers will remain a profitable and demanding subject for academic and clinical settings as well as industry. Constant improvements to this PLGA polymeric nanoformulation, along with the researchers’ encyclopedic expertise, will revolutionize how we approach illness detection and treatment, with a special focus on enhancing patients’ quality of life.

## Figures and Tables

**Figure 1 pharmaceutics-14-02728-f001:**
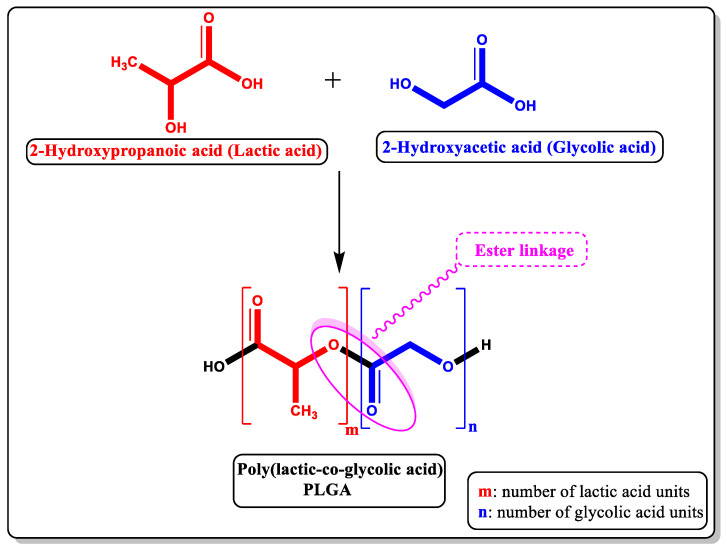
Schematic representation of chemical structure for PLGA and its monomers. Due to the presence of ester linkages that are degraded by hydrolysis in aqueous environments.

**Figure 2 pharmaceutics-14-02728-f002:**
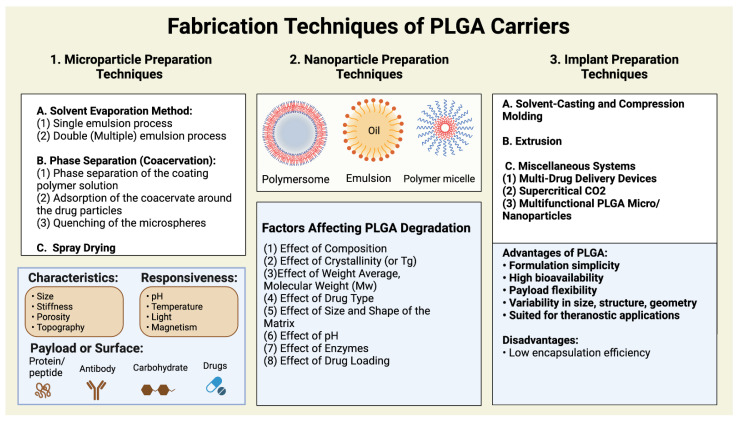
The figure shows the different and most important fabrication techniques of PLGA, and the factors related to its degradation besides characteristics, responsiveness, advantages, and disadvantages. Due to the biodegradability and biocompatibility of PLGA polymer, it is widely used as a hydrophobic polymer to form various kinds of carriers such as polymersomes, emulsions, and polymeric micelles, and many more. Also, the figure shows some important PLGA characteristics, responsiveness, factors affecting its degradation, and some advantages and disadvantages of utilizing PLGA as a carrier system. The figure was created with BioRender.com.

**Figure 3 pharmaceutics-14-02728-f003:**
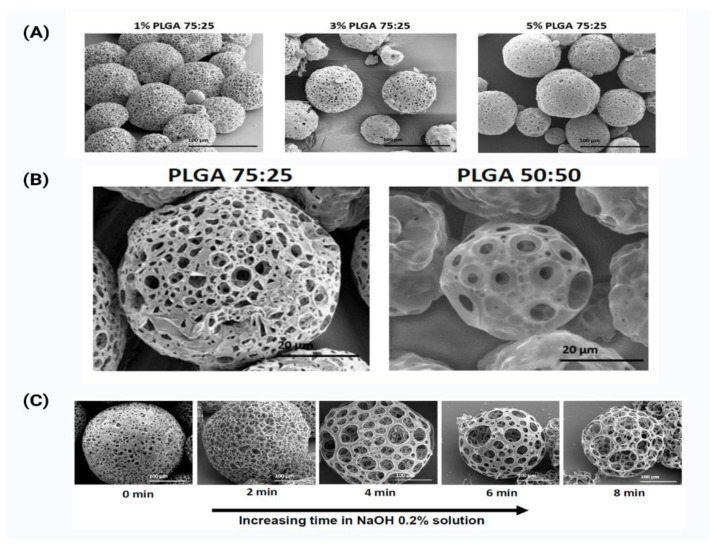
The figure shows different scanning electron microscope (SEM) images of poly(lactic-acid (PLGA) materials. (**A**) PLGA porous microsphere made by batch synthesis with increasing PLGA 75:25 concentrations (%*w/v*), 1%, 3%, and 5%. (**B**) SEM images representing the morphology of porous and non-porous microspheres obtained with PLGA polymers and MWT using PLGA 75:25, PLGA 50:50. (**C**) SEM images of 3% (*w/v*) PLGA 75:25 porous microspheres taken by the microfluidic technique at increasing soaking time in NaOH 0.2% solution. As immersion time increased, a more porous structure was obtained. This figure is adapted from [30].

**Figure 4 pharmaceutics-14-02728-f004:**
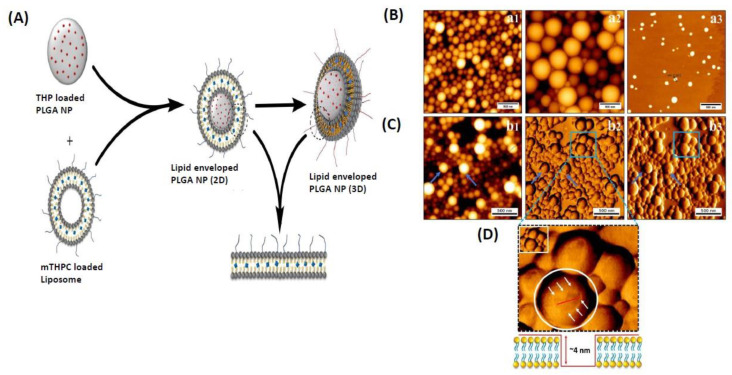
Biodegradable nanoparticles with lipid bilayers are shown schematically in (**A**). Morphology of the surface depicted in (**B**–**D**). (**A**) AFM imaging of THP-NP 200 nm, THP-NP 400 nm, and mTHPC-LP. AFM micrographs of lipid-coated nanoparticles are shown in panel (**B**). (a1) THP NP 200 nm; (a2) THP NP 400 nm and (a3) mTHPC-LP using AFM. Lock-in amplitude view, lock-in phase view, and height measured view. Lipoparticle thickness around PLGA nanoparticles is seen in (**C**) using an AFM cross-sectional profile. (b1) height measured view; (b2) lock-in amplitude view and (b3) lock-in phase view. The bar’s width corresponds to 500 nm. The figure is adapted from [71] with copyright permission.

**Figure 5 pharmaceutics-14-02728-f005:**
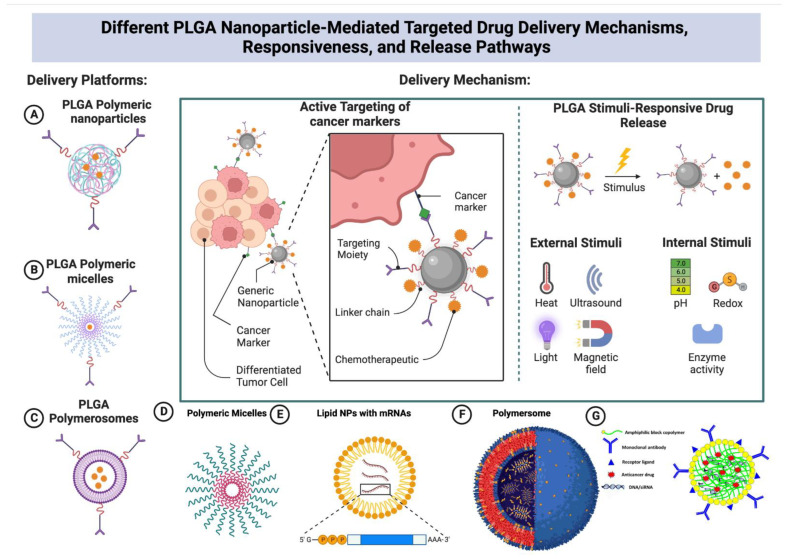
The images depict different PLGA polymer-based carriers that have been fabricated over recent years with the most important types used especially for targeting cancer (Figures (**A**–**F**)). Also, Figure (**G**) shows how PLGA amphiphilic block copolymers looks when loading with different anticancer drugs and/or DNA or RNA and decorated with small molecules receptor ligands or monoclonal antibodies for active targeting. The active targeting of cancer utilizing PLGA functionalization with different cancer markers makes them one of the most promising platforms for both treatment and diagnostics. The PLGA stimuli responsive release could be external or internal stimuli. The figure was created with BioRender.com.

**Figure 6 pharmaceutics-14-02728-f006:**
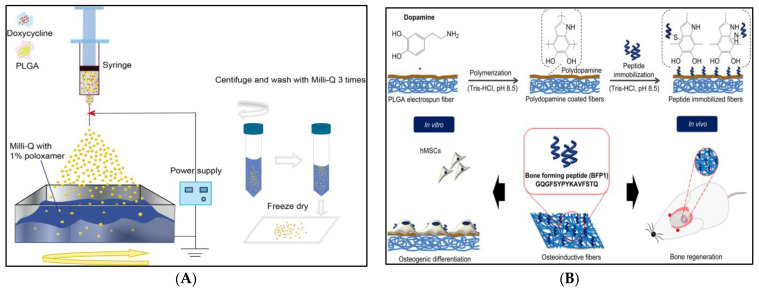
The figure represents different applications of PLGA. (**A**) Schematic diagram of the fabrication of drug-loaded PLGA microspheres by electrospraying. (**B**) Schematic illustration of the preparation of Dopamine with immobilized BFP1 by polydopamine coating. Figure (**A**) is reproduced from [33] and Figure (**B**) from [94] with copyright permission.

**Figure 7 pharmaceutics-14-02728-f007:**
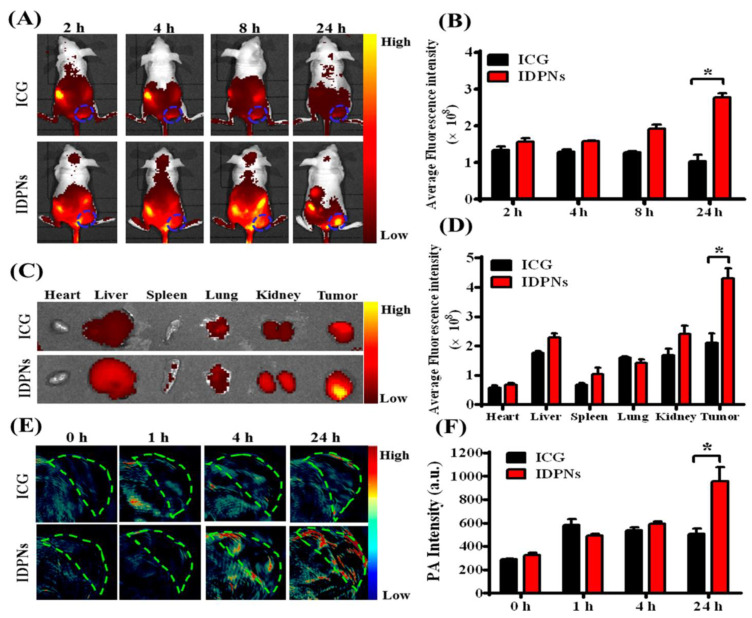
(**A**) Time-dependent in vivo whole-body near-infrared (NIR) fluorescence (FL) imaging of EMT-6 tumor-bearing BALB/c nude mice 2, 4, 8, and 24 h following i.v. administration of free ICG and IDPNs. Tumors are blue circles. The color bar gradually changes from red to yellow as fluorescence signal intensity increases. (**B**) Quantification of average fluorescence signals in tumor sites in part A. * *p* < 0.05. (**C**) The color bar gradually changes from red to yellow as fluorescence signal intensity increases. (**D**) Quantification of part C’s isolated organs and tumors’ average fluorescence signals. * *p* < 0.05. (**E**) Time-dependent in vivo photoacoustic (PA) imaging of EMT-6 tumors in BALB/c nude mice 0, 1, 4, and 24 h following i.v. administration of free ICG and IDPNs. Tumor locations are indicated by cycles. The color bar gradually changes from blue to red as PA intensity rises. (**F**) PA intensity quantification in part E tumor locations. * *p* < 0.05. The figure is adapted from [99] with copyright permission.

**Figure 8 pharmaceutics-14-02728-f008:**
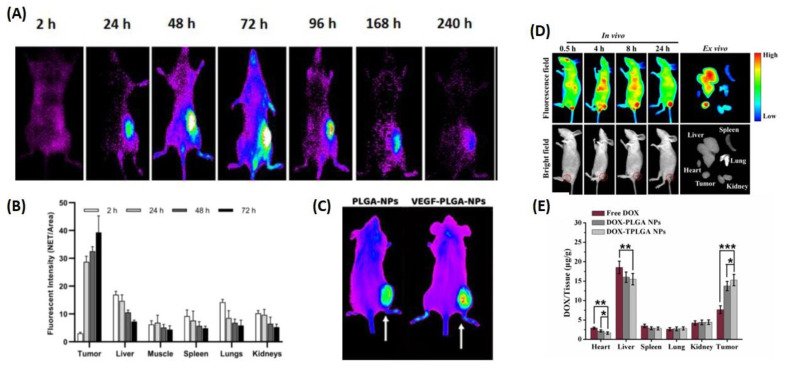
The figure represents different applications of PLGA. (**A**) Optical photographs taken of the entire mouse after 2, 24, 48, 72, 96, 168, and 240 h after subcutaneous injection of 500 µg of fluorescent PLGA-NPs in the right flank in a syngeneic tumor-bearing mouse. (**B**) Analyzing the PLGA-NPs’ biodistribution in BALB/c mice. This data is shown as the mean fluorescence (NET/Area) ±SD from 5 different mice at each time point. (**C**) Images of two syngeneic J744a-carrying mice. 1 tumor located in the right thigh, obtained 24 h post-injection (p.i.) of either native PLGA-NPs (left) or VEGF-PLGA-NPs (right). (**D**) TPLGA NP biodistribution in vivo. 24 h post-injection ex vivo fluorescent pictures of key organs and 4T1 tumors were observed. Red circles indicate tumor locations. Fluorescence intensity decreases from red to blue. (**E**) Tumor and major organ DOX content was determined 24 h after tail-vein injection with free DOX and DOX-loaded NPs. Mean ± SD (*n* = 3) was used. * *p* < 0.05, ** *p* < 0.01, and *** *p* < 0.001. Figure (**A**–**C**) is reproduced from [101] and Figure (**D**,**E**) are adapted from [96] with copyright permission.

**Table 1 pharmaceutics-14-02728-t001:** The table represents different fabrication methods, in addition to describing the process, main advantage, and negative aspects. All information was taken from the following references [8,59,60].

Method	Main Idea	Main Advantage	Negative Aspects
1-Nanoprecipitation method	The nanoprecipitation method produces lipid-polymer particles with a high yield of less than 100 nanometers.	Nanoparticles with a higher production rate and better size homogeneity.	Organic solvents have the potential to damage biomolecules (protein nuclei acids).
2-Microfluidic method	This method can control immiscible liquids in small quantities in a precise capillary network with microscale fluid channels.	Small size distribution (lower PDI), higher encapsulation and loading efficiencies, improved batch-to-batch uniformity, and simple scale-up possibilities all contribute to cost-effective nanocarrier development.	The yield Is relatively poor.
3-One-step method	This approach uses covalent conjugation to combine various lipid and polymer precursors.	Cost-effectiveness, scalability, and a traditional method of preparation.	Instability of biomacromolecules.
4-Two-step method	Monolayer, bilayer, and multilayer shells are usually made with it. Cationic lipid vesicles are coupled with anionic polymeric nanoparticles using electrostatic interactions in this process.	Nanoparticles produced easily cross the membrane barrier and circulate in the bloodstream, allowing them to deliver drugs for long periods of time.	Separate preparation of polymeric nanoparticles and lipid vesicles, which takes a long time and consumes a lot of resources.
5-Spray-drying method	Polymers are used to make nanoparticles with sizes ranging from 400 to 500 nm, which are then dispersed in an organic solvent containing various lipids. To complete the product, the lipoidal polymeric suspension is spray dried.	Fast and effective. Ideally suited to commercial scale-up. Protein parameters that are more appropriate.	Small lots with a moderate yield.

**Table 2 pharmaceutics-14-02728-t002:** Some examples of preclinical studies of PLGA-based nanomaterial therapy that has been used as anticancer delivery systems.

Nanoparticle (NPs)	Polymer and Additives	Function of Polymer	Drug	Type of Cancer	Type of Cell Line	Target Action	Reference
Afatinib-loaded PLGA NPs)	PLGA	Protect Afatinib, improve drug delivery	Afatinib	Colon Cancer	Caco-2 cells	pH-responsive characteristics to increase the sensitivity of colon cancer cells to afatinib.	2019 [84]
Platinum–curcumin complexes loaded into pH and redox dual-responsive nanoparticles (PteCUR@ PSPPN)	mPEG-SS-PBAE-PLGA	Control intracellular release, synergistic anticancer effects	Platinum–curcumin	Lung Cancer	A549 cells	Synergistic anticancer effects, enhanced anti-metastatic activity	2019 [85]
Uncaria tomentosa extract (UT)-PLGA & UTPCL	PCL and PLGA	Better drug delivery—UT-PLGA nanoparticles showed higher drug loading	Uncaria tomentosa extract	Prostate Cancer	LNCaP, DU145 cells	UT-PLGA showed higher cytotoxicity towards DU145 cells, UTPCL showed higher cytotoxicity against LNCaP cells	2019 [86]
Curcumin (Cur)-loaded polymeric poly (lactic-co-glycolic acid) (PLGA) nanoparticles (Cur-PLGA NPs)	PLGA	Stabilize curcumin in the presence of light, improved serum stability compared with free curcumin	Curcumin	Ovarian Cancer	SKOV3 cells	Cytotoxic effects on tumor cells upon irradiation at a low intensity inhibits tumor growth	2019 [87]
5-FU-Chrysin-loaded PLGA-PEG-PLGA nanoparticles (5FU-Chrysin-PLGA-PEG-PLGA NPs)	PLGA-PEG-PLGA	Improve the functional delivery efficacy of 5-FU and Chrysin in cancer	5-FU, Chrysin	Colon Cancer	HT-29 cells	Apoptosis, growth inhibitory effects	2020 [88]
Sorafenib (SF)-loaded catatonically-modified polymeric nanoparticles (NPs)	PLGA	Aerosolization efficiency for pulmonary delivery	Sorafenib	Lung Cancer	A549 cells	Enhanced cell migration inhibition, reduction in cell survival, inhibition in the formation of colonies	2020 [89]

**Table 3 pharmaceutics-14-02728-t003:** Selected clinical trials enlisted in clinicaltrials.gov website for marketed PLGA-based therapeutics. Reproduced from [10].

Brand Name	Indication	Clinicaltrials.Gov Identifier
Somatuline^®^ LA	Acromegaly	NCT03066245
Sandostatin^®^ LAR	Acromegaly and carcinoid	NCT03060655
Nutropin Depot ^®^	Growth deficiency	NCT02568527
Zoladex^®^	Breast cancer. Prostate cancer	NCT03474900
Arestin	Periodontal disease	NCT02726646
Trelstar^TM^ Depot	Advanced Prostatic Cancer	NCT01681381
Suprecur^®^ MP	Prostate cancer	NCT0 1753 089
Pamorelin^®^	Prostate cancer	NCT03045913
Lupron Depot	Prostate cancer	NCT02578069
Eligard	Advanced Prostatic Cancer	NCT03401216
Atridox^®^	Chronic adult periodontitis	NCT03429803
Risperidal^®^ Consta	Antipsychotic	NA
Decapepty	Prostate cancer	NA

**Table 4 pharmaceutics-14-02728-t004:** Current clinical trials enlisted in clinicaltrials.gov website. ClinicalTrials.gov search results on 30 July 2022.

Clinical Trial No. (Status)	Study Title	Conditions	Interventions
NCT05475444 (Completed)	PLGA Nanoparticles Entrapping Ciprofloxacin to Treat E-Fecalis Infections in Endodontics	Bacterial Infections Oral	Device: Chitosan-coated PLGA nanoparticles entrapping Ciprofloxacin incorporated in smart gels Device: Ciprofloxacin paste and solution
NCT03060655 (Unknown status)	Study of PLGA-Mg Material in Clinical Orthopedics	Fracture Dislocation	Biological: PLGA-Mg material Biological: titanium alloy
NCT04735601 (Recruiting)	Ahmed Valve Implantation Coated With Poly Lactic-Co-glycolic Acid (PLGA) Saturated With Mitomycin-C in the Management of Adult Onset Glaucoma in Sturge Weber Syndrome	Glaucoma	Procedure: Ahmed Valve
NCT03066245 (Unknown status)	Use of Stem Cells Cultured on a Scaffold for the Treatment of Aneurysmal Bone Cysts (ABC)	Aneurysmal Bone Cyst	Biological: MSC-PLGA
NCT03474627 (Completed)	PLGA-coated Biphasic Calcium Phosphate in Sinus Lift	Sinus Floor Augmentation	Device: Non-coated HA/TCP particles Device: PLGA-coated HA/TCP particles
NCT02487186 (Completed)	Locally Delivered Doxycycline Adjunct to Nonsurgical Periodontal Therapy.	Periodontal Disease	Drug: Doxycycline Procedure: Full-mouth debridement
NCT02568527 (Completed)	Biodegradable Synthetic Scaffold as a Substitute for hAM in Limbal Epithelial Cells Transplant in LSCD Patients	Limbal Stem Cell Deficiency	Device: PLGA scaffold
NCT00836797 (Completed)	Radiographic Assessment of Bone Regeneration in Alveolar Sockets with PLGA Scaffold	Preservation of Alveolar Bone Height With PLGA Bioscaffold	NA
NCT04619056 (Recruiting)	First-in-man Clinical Trial of CEB-01 PLGA Membrane in Recurrent or Locally Advanced Retroperitoneal Soft Tissue Sarcoma	Locally Advanced Soft Tissue Sarcoma Recurrent Soft Tissue Sarcoma	Combination Product: CEB-01 membrane loaded with SN-38
NCT04848818 (Recruiting)	Comparative Trial of Operative Treatment of Distal Pediatric Forearm Fractures With Biodegradable Nails and K-wires	Fracture Wrist	Procedure: Distal radial metaphyseal fracture fixation with percutaneous K-wires Procedure: Distal radial and/or ulnar metaphyseal fracture fixation with biodegradable PLGA-based (Activa Im-Nail) implants
NCT05442736 (Completed)	Modified Surface of PLGA Nanoparticles in Smart Hydrogel	Antibiotic Resistant Infection	Drug: Ciprofloxacin
NCT03474900 (Completed)	Biodegradable Versus Titanium Nailing in Forearm Shaft Fractures in Children	Forearm Fracture	Device: PLGA implant, Bioretec ltd. Finland Device: Titanium elastic stable nail
NCT05456022 (Not yet recruiting)	Therapeutic Efficacy of Quercetin Versus Its Encapsulated Nanoparticle on Tongue Squamous Cell Carcinoma Cell Line	Oral Cancer	Drug: Quercetin 3,3’,4’,5,6-Pentahydroxyflavone, 2-(3,4-Dihydroxyphenyl)-3,5,7-trihydroxy-4H-1-benzopyran-4-one Drug: Quercetin-encapsulated PLGA-PEG nanoparticles (Nano-QUT) Drug: Doxorubicin chemotherapeutic drug as a positive control
NCT01729195 (Completed)	Ankle Syndesmosis Fixation by Antibiotic Releasing Bioabsorbable Screw	Ankle Fracture	Procedure: A ciprofloxacin containing bioabsorbable PLGA bone screw
NCT02726646 (Completed)	Evaluation of Local Doxycycline in Smokers With Chronic Periodontitis	Chronic Periodontitis	Drug: Doxycycline Procedure: Mechanical debridement Drug: Placebo
NCT04339764 (Recruiting)	Autologous Transplantation of Induced Pluripotent Stem Cell-Derived Retinal Pigment Epithelium for Geographic Atrophy Associated With Age-Related Macular Degeneration	Age-Related Macular Degeneration	Drug: iPSC-derived RPE/PLGA transplantation
NCT01681381 (Unknown status)	Evaluate Safety and Effectiveness of the Tivoli^®^ DES and The Firebird2^®^ DES for Treatment Coronary Revascularization	Ischemic Heart Disease Myocardial Ischemia Coronary Artery Lesions, Primary Coronary Disease Acute Coronary Syndrome Furcation Lesion of Coronary Artery	Device: Tivoli^®^ DES Device: Firebird2^®^ DES
NCT02017275 (Completed)	Comparison of BuMA eG Based Biodegradable Polymer Stent with EXCEL Biodegradable Polymer Sirolimus-eluting Stent in “Real-World” Practice	Coronary Artery Disease	Device: BuMA Device: EXCEL
NCT01753089 (Active, not recruiting)	Dendritic Cell Activating Scaffold in Melanoma	Melanoma	Biological: WDVAX
NCT04751786 (Recruiting)	Dose Escalation Study of Immunomodulatory Nanoparticles	Advanced Solid Tumor	Drug: PRECIOUS-01
NCT04941612 (Recruiting)	Use of the Bioabsorbable Activa IM-Nail™ in Pediatric Diaphyseal Forearm Fractures	Fracture Fixation, Intramedullary Forearm Fracture Fracture Healing Child, Only Implant Complication	Device: Activa IM-Nail
NCT04385745 (Unknown status)	Treatment of Children’s Forearm Shaft Fractures With Biodegradable Intramedullary Nailing, Compared With Elastic Stable Intramedullary Nailing	Fractures, Bone Injury Arm	Procedure: Biodegradable Intramedullary Nailing (BIN)
NCT02255188 (Completed)	Experimental Study of the Vascular Prosthesis Manufactured by Electrospinning	Arterial Occlusive Disease	Procedure: blood sampling procedure
NCT03707769 (Recruiting)	TIPS Microspheres for Perianal Fistula	Perianal Fistula	Device: TIPS microspheres
NCT05448625 (Recruiting)	DES in Patients With a High Risk of Ischemic Events	Drug-eluting Stent Coronary Artery Disease	Device: Genoss DES
NCT03045913 (Active, not recruiting)	Genoss DES Prospective Multicenter Registry	Coronary Artery Disease Myocardial Ischemia Myocardial Infarction	Device: Genoss DES
NCT04082962 (Recruiting)	Dexamethasone Implant for Retinal Detachment in Uveal Melanoma	Exudative Retinal Detachment Uveal Melanoma	Drug: Dexamethasone intravitreal implant
NCT03762655 (Active, not recruiting)	Study of Probable Benefit of the Neuro-Spinal Scaffold™ in Subjects with Complete Thoracic AIS A Spinal Cord Injury as Compared to Standard of Care	Injury, Spinal Cord	Device: Neuro-Spinal Scaffold
NCT04094298 (Recruiting)	Use of Extended Release Triamcinolone in the Treatment of Rotator Cuff Disease	Rotator Cuff Tears Rotator Cuff Tendinitis Rotator Cuff Impingement	Drug: FX006 Injection Injections, Glucocorticoids
NCT05104853 (Recruiting)	Study to Evaluate the Safety, Tolerability, PDs, and Efficacy of CNP-104 in Subjects With Primary Biliary Cholangitis	Primary Biliary Cholangitis	Drug: CNP-104 Drug: Placebo
NCT05250856 (Recruiting)	CNP-201 in Subjects with Peanut Allergy	Peanut Allergy	Drug: CNP-201 Drug: Placebo
NCT02578069 (Completed)	First-in-man Trial Evaluating the Safety and Efficacy of the NOVA Intracranial Stent (NOVA Trial)	Ischemic Stroke	Device: NOVA Intracranial Sirolimus Eluting Stent System Device: Apollo Intracranial Stent System
NCT04012567 (Active, not recruiting)	Safety and Effectiveness of BIOSURE RG in Cruciate Ligaments Reconstruction in Chinese	Cruciate Ligament Reconstruction, Knee	Investigational device: Biosure Regenesorb Interference Screw Control device: Biosure HA Interference Screw
NCT02982889 (Completed)	Single Ascending Dose Study of Two Liquidia Bupivacaine Formulations	Acute Pain	Drug: LIQ865A bupivacaine formulation Drug: LIQ865B bupivacaine formulation Drug: Diluent for LIQ865 Drug: 0.5% bupivacaine hydrochloride
NCT03401216 (Unknown status)	Stent Coverage and Neointimal Tissue Characterization After Extra Long Everolimus—Eluting Stent Implantation	Ischemic Heart Disease Coronary Artery Disease Coronary Atherosclerosis	Device: SYNERGY 48 mm Procedure: PCI Procedure: 3-month OCT follow-up Procedure: 6-month OCT follow-up
NCT01734512 (Active, not recruiting)	Phase II Study of Everolimus for Recurrent or Progressive Low-grade Gliomas in Children	Pediatric Recurrent Progressive Low-grade Gliomas Pediatric Progressive Low-grade Gliomas	Drug: Everolimus
NCT03429803 (Active, not recruiting)	DAY101 In Gliomas and Other Tumors	Low-grade Glioma	Drug: DAY101

NCT = Clinical trial numbers; NA = not available.

## Data Availability

Not applicable.

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
