# Peer review of "PLGA-Based Nanomedicine: History of Advancement and Development in Clinical Applications of Multiple Diseases"

_pharmaceutics, 2022, doi:10.3390/pharmaceutics14122728_

Round 1

Reviewer 1 Report

Alsaab et al. presented a Review article for Pharmaceutics titled “PLGA-Based Nanomedicine: History of Advancement and Development in Clinical Applications of Multiple Diseases”. In this manuscript the authors systematically described recent research on PLGA-based nanosystems for several diseases. Given the vastness of the subject, not focused on a single disease, the manuscript needs a substantial reorganization to avoid confusion in the reader and other MAJOR REVISIONS before to be considered for publication.

 Specific comments:

-        Nanoparticles is sometimes written in capital letter, such as in Abstract section, please revise it.

-        Studies should not be reported as a mere list but they should include more details about methods and obtained results.

-        The composition and mechanism of each drug delivery systems can be better detailed. Such an example in 3.1.2. section liposomes are mentioned for the first time without any description of the carrier.

-        Since different drug delivery technologies have been reported, conclusions of each section as well as comparisons between different nanotechnologies with pros/cons could be included.

-        Representative figures from the cited papers should be added in the manuscript to enhance the comprehension of the work.

-        Tables should have a highly communicative role and should be comprehensible also when detached from the main text. Table 1 caption should be improved as the subject is missing. Moreover, only two references are continuously repeated and they can be included in the caption.

-        Table 3 mentions different ongoing clinical trials in which PLGA-based nanoparticles are involved, considering several diseases. However, authors only deepen preclinical studies on four macro-areas neglecting others. New sections focusing on ocular diseases, periodontal diseases, pulmonary diseases and fractures should be added. Please consider the following manuscript and others:

Int. J. Mol. Sci. 2018, 19(9), 2830; https://doi.org/10.3390/ijms19092830

Molecules 2021, 26(6), 1643; https://doi.org/10.3390/molecules26061643

BioMed Research International 2022, Article ID 5058121; https://doi.org/10.1155/2022/5058121

-        Could you please provide the permission for the figures realised with BioRender?

-        Section 4. Only Lupron® has been reported by the authors but other drugs have been approved by FDA such as Zoladex Depot ®(AstraZeneca UK Limited) or Sandostatin Lab ®. Please, update this section.

Author Response

Dear/ Respected reviewers

Thank you for giving us the opportunity to resubmit a revised draft of our manuscript titled [PLGA-Based Nanomedicine: History of Advancement and Development in Clinical Applications of Multiple Diseases], manuscript ID: pharmaceutics-1953533.

I and my co-authors appreciate the time and effort that you and the reviewers have dedicated to providing your valuable and positive feedback on our manuscript. We are grateful to the reviewers for their insightful comments on our paper. We tried as much as possible to respond to most of the enquiries and suggestions provided by the respected reviewers. All changes were made through Microsoft word track changes. Here is our point-by-point response to the reviewers’ comments and concerns followed by references to some responses.

Accept our regards.

Reviewer 1: Comments and Suggestions for Authors

 Alsaab et al. presented a Review article for Pharmaceutics titled “PLGA-Based Nanomedicine: History of Advancement and Development in Clinical Applications of Multiple Diseases”. In this manuscript the authors systematically described recent research on PLGA-based nanosystems for several diseases. Given the vastness of the subject, not focused on a single disease, the manuscript needs a substantial reorganization to avoid confusion in the reader and other MAJOR REVISIONS before to be considered for publication.

 Specific comments:

1-        Nanoparticles is sometimes written in capital letter, such as in Abstract section, please revise it.

Answer: The manuscript has been revised for English language clarity and modifying all grammatical errors including this comment.

2-        Studies should not be reported as a mere list but they should include more details about methods and obtained results.

Answer: We have modified and added some more details in each section as suggested.

3-        The composition and mechanism of each drug delivery systems can be better detailed. Such an example in 3.1.2. section liposomes are mentioned for the first time without any description of the carrier.

Answer: We have modified and added some more details in 3.1.2. section as suggested.

4-       Since different drug delivery technologies have been reported, conclusions of each section as well as comparisons between different nanotechnologies with pros/cons could be included.

Answer: We have modified and added some more details in some section as suggested.

5-        Representative figures from the cited papers should be added in the manuscript to enhance the comprehension of the work.

Answer:  We have added 2 More figures to the total from cited references. So, we have 7 figures and 4 tables.

6-        Tables should have a highly communicative role and should be comprehensible also when detached from the main text. Table 1 caption should be improved as the subject is missing. Moreover, only two references are continuously repeated, and they can be included in the caption.

Answer: We have modified Table 1 and cited the references in table caption as suggested.

7-        Table 3 mentions different ongoing clinical trials in which PLGA-based nanoparticles are involved, considering several diseases. However, authors only deepen preclinical studies on four macro-areas neglecting others. New sections focusing on ocular diseases, periodontal diseases, pulmonary diseases and fractures should be added. Please consider the following manuscript and others:

Int. J. Mol. Sci. 2018, 19(9), 2830; https://doi.org/10.3390/ijms19092830

Molecules 2021, 26(6), 1643; https://doi.org/10.3390/molecules26061643

BioMed Research International 2022, Article ID 5058121; https://doi.org/10.1155/2022/5058121

Answer: We have added a new section 6.5 (6.5.1., 6.5.2., 6.5.3., and 6.5.4.) about other diseases and included ocular diseases, periodontal diseases, and pulmonary diseases with some selected examples for different uses of PLGA in these diseases and we have cited these references as required. 

8- Could you please provide the permission for the figures realised with BioRender?

Answer: We would like to thank the reviewer for the comment.  We have provided the journal with all created figures from BioRender or taken from other sources.

9- Section 4. Only Lupron® has been reported by the authors but other drugs have been approved by FDA such as Zoladex Depot ®(AstraZeneca UK Limited) or Sandostatin Lab ®. Please, update this section

Answer: Thanks for the reviewer for  the comment. There are many other US FDA approved PLGA based marketed products such as Zoladex, Sandostatin® LAR, Suprecur® MP, and many other more as listed in Table 3.

Reviewer 2 Report

This manuscript is written clearly and intelligibly. The experimental results are consistent with the conclusions made by the authors. I consider the content of this manuscript will be of some interest to readers. However, the authors should improve the soundness of this study.

It might be interesting for the authors to familiarize themselves with the work of https://www.mdpi.com/1996-1944/14/22/6915, where similar studies were carried out.

Author Response

This manuscript is written clearly and intelligibly. The experimental results are consistent with the conclusions made by the authors. I consider the content of this manuscript will be of some interest to readers. However, the authors should improve the soundness of this study.

It might be interesting for the authors to familiarize themselves with the work of https://www.mdpi.com/1996-1944/14/22/6915, where similar studies were carried out.

Answer:  we would like to thank the reviewer for the insightful comment. We have checked and cited the mentioned reference as it is a great addition to our review. We have tried as much as possible to improve our manuscript as suggested.

Round 2

Reviewer 1 Report

Authors adequately modified the manuscript.